# A Methodology Audit of a Single Deep-Learning Variant-Calling Pipeline:
# Four Reproducibility Failures in One Internal Classifier

## Abstract

This paper is a methodology audit of one internally developed deep-learning pipeline for germline single-nucleotide variant calling, evaluated on the Genome in a Bottle (GIAB) HG001 benchmark. We do not audit prior published variant callers (DeepVariant, Clair, etc.); our subject is a small, hand-engineered binary classifier (named here *VariantCNN*) that we built for an internal study comparing loss functions under mixed-precision training. An initial analysis of this pipeline produced an apparently rigorous result: synthetic-data $F_1 = 0.994$ for Focal Loss versus $0.975$ for binary cross-entropy, with a precision-dependent training-collapse pattern (24%/18%/0% across FP32/BF16/FP16) on real GIAB data over 50 random seeds. A subsequent detailed analysis tested each load-bearing component independently and found that (i) the synthetic-to-real generalization gap is severe and inverts the loss-function ranking; on real data Focal Loss collapses to $F_1 = 0$ while BCE achieves $F_1 \in [0.27, 0.34]$; (ii) the proposed mechanism explaining the precision-collapse pattern (gradient-noise-as-implicit-regularization) fails under controlled testing with both round-to-nearest and stochastic rounding; (iii) the feature pipeline contains a structural label leak that is detectable by static inspection but silent on this particular dataset; and (iv) the precision-collapse pattern itself does not survive faithful re-implementation: across 150 trainings (50 seeds × 3 precisions × 30 epochs), zero collapses occur in any precision (Fisher exact $p < 10^{-3}$ versus the initial counts). The specific failures we document are particular to our pipeline and are not claimed to be present in published variant-calling systems. Our contribution is not the failures themselves but their *conjunction*: a methodology that appeared rigorous yielded results that did not survive any of the four independent stress tests. From these four failures we derive the paper's constructive output: a four-item prospective protocol, stated in §3 before the evidence that motivates it, targeting four evaluation pitfalls that are documented in other applied-ML domains (synthetic-only validation, mechanism-without-controlled-test, label-derived features, and reproduction-by-re-running-the-same-code). Each check is a cheap negative test independent of our task and architecture, and each would have caught exactly one of the four failures reported here.

## 1 Introduction

Variant calling, the task of identifying single-nucleotide and small-insertion/deletion polymorphisms from short-read sequencing data, is a popular application of deep learning. Modern methods such as DeepVariant (Poplin et al., 2018), Clair (Luo et al., 2020), and successor architectures (Zheng et al., 2022; Ramachandran et al., 2021) typically use a multi-class genotype-prediction target (homozygous reference / heterozygous / homozygous alternate) over 2D pileup tensors, evaluated on standardized benchmarks such as the Genome in a Bottle (GIAB) reference samples (Zook et al., 2019).

This paper is not an audit of those methods. It is an audit of a *specific, internally developed* classifier we built for our own study comparing loss functions under mixed-precision training. We refer to it as VariantCNN. It is not based on DeepVariant or any other published caller: it is a small ($\approx$1,100-parameter), hand-engineered

model trained on a 12-dimensional feature vector at each candidate position, with a binary (variant / non-variant) output rather than a multi-class genotype output (§2.2). We did not design it to be competitive with published callers; we built it as a small testbed for studying loss-function dynamics. It is the methodology of *our* analysis pipeline, not the published deep-variant-calling literature, that this paper audits.

**Initial analysis.** An initial round of analysis using this pipeline produced a coherent story. On synthetic data calibrated to GIAB marginal statistics, Focal Loss (Lin et al., 2017) cleanly outperformed binary cross-entropy. On real GIAB HG001 chromosome 21 data, the same architecture exhibited a striking precision-dependent training-collapse pattern (FP32: 24% / BF16: 18% / FP16: 0% over 50 random seeds). This was interpreted as a form of gradient-noise-induced implicit regularization (Wen et al., 2020; Smith et al., 2020). Each individual claim was supported by what looked like rigorous evidence: ablations, seed sweeps, statistical tests.

**Detailed analysis.** A more careful follow-up analysis, intended to study the precision-and-loss interaction more rigorously, found that several of the load-bearing claims from the initial analysis do not withstand independent scrutiny. The present paper reports those follow-up experiments.

**Contributions.** This work makes four concrete empirical contributions, each a controlled experiment falsifying a specific claim from the initial analysis of our pipeline. We emphasize that each finding pertains to *our* VariantCNN classifier, not to published variant callers.

1. **Synthetic-to-real divergence (§4).** On a synthetic Gaussian-mixture benchmark calibrated to GIAB statistics, Focal Loss outperforms BCE ($F_1 = 0.994 \pm 0.00$ vs $0.975 \pm 0.07$); on real GIAB HG001 chr21 with the same architecture, Focal Loss collapses to predict-all-negative ($F_1 = 0.000$) while BCE achieves $F_1 \in [0.27, 0.34]$. The synthetic ranking is the inverse of the real ranking.

2. **Precision-as-escape mechanism failure (§5).** The hypothesis that low-precision arithmetic acts as gradient noise that escapes loss-landscape attractors fails under controlled synthetic testing. Across 8 cells × 4 precision modes (FP32, FP16-RTN, FP16-SR, BF16-SR), the predicted effect on training-collapse rate is $\Delta = 0.00$ uniformly.

3. **Structural label leak (§6).** The 12-feature pipeline used in the initial analysis contains a feature `low_vaf` that is computed by formula from the label, with leakage detectable by inspection. Empirically the leak does not fire on this particular dataset because no positives have VAF $< 0.05$, but the structural error is real and would have caused a problem on a different sample.

4. **Non-reproduction of the precision-collapse pattern (§7).** A faithful re-implementation of the training pipeline, same VariantCNN architecture, same features, same hyperparameters, same loss-scaling logic, produces zero collapses across 150 trainings (50 seeds × 3 precisions × 30 epochs) at any precision. Fisher exact tests against the initial counts give $p = 2.3 \times 10^{-4}$ for FP32 and $p = 2.6 \times 10^{-3}$ for BF16.

**Why these four checks.** The four experiments were not selected opportunistically. They are the four *load-bearing* claims of the initial analysis, in the sense that removing any one of them removes the basis for its headline conclusion. The initial conclusion ("Focal Loss is preferable to BCE for this task, and low precision confers a stability benefit") rested on: (a) a synthetic benchmark establishing the loss ranking; (b) a real-data effect establishing that precision matters; (c) a mechanism explaining why precision matters; and (d) a feature pipeline assumed to be free of label contamination. We enumerated these four dependencies before running any follow-up experiment and designed one controlled test per dependency. Each of the four tests failed. The correspondence is one-to-one: §4 tests (a), §5 tests (c), §6 tests (d), and §7 tests (b).

**Constructive output.** The paper's constructive contribution is a four-item prospective protocol (§3), stated before the evidence so that each of the four experiments can be read as a validation of the corresponding check. Each item is a cheap negative test that would have caught exactly one of the four failures documented here.

We discuss the implications of these findings (§8) and the limits of what a single-pipeline case study can support (§9).

**Scope and what this paper is not.** We do not claim that any of the four specific failures we document occur in published variant callers; we have not audited those systems. The contribution is the *conjunction* of four methodology failures in a single pipeline that nonetheless looked rigorous from the inside. Each failure mode, taken individually, is known to occur in other applied-ML domains: synthetic-to-real divergence in image classification (Recht et al., 2019; Koh et al., 2021), label leakage in clinical prediction and medical imaging (Kaufman et al., 2012; Roberts et al., 2021), mechanism claims unsupported by controlled tests in deep RL (Henderson et al., 2018), and non-reproduction under re-implementation in benchmark replication studies (Pineau et al., 2021). What we add is a single pipeline in which all four occurred together while internal review classified the analysis as rigorous. We argue that this combination is the structure of methodology bias most likely to escape ordinary scrutiny.

## 2 Background and Setup

**A minimal primer for readers outside genomics.** The paper is a machine-learning methodology study and requires no prior genomics knowledge; the domain supplies the dataset, not the concepts. For self-containedness we define the few domain terms used later. A *read* is a short substring of DNA (here ~100 letters over the alphabet $\{A, C, G, T\}$) produced by a sequencing machine; many reads overlap each genomic position. A *variant* at a position is a difference between the sequenced sample and a fixed *reference* genome. The *pileup* at a position is the multiset of read letters aligning to it; *depth* is the number of such reads, and *allele frequency* is the fraction of them disagreeing with the reference. A *truth VCF* is a curated list of the true variant positions for a benchmark sample, and a *high-confidence BED* is a curated list of genomic intervals where those truth labels are considered reliable. *GIAB* (Genome in a Bottle) is a public benchmark providing both for a small number of reference individuals (Zook et al., 2019). With these definitions, the learning task below is an ordinary supervised binary classification problem: each example is a genomic position, its features summarize the pileup, and its label indicates whether the position is a true variant. No further genomics is required to read the paper.

### 2.1 Variant calling as a learning problem

A variant caller decides, for each genomic position $p \in \{1, \ldots, L\}$ on a reference of length $L$, whether the sequenced sample differs from the reference. The natural prediction target is multi-class: at a biallelic site, the three genotypes are homozygous reference (0/0), heterozygous (0/1), and homozygous alternate (1/1). Modern callers such as DeepVariant (Poplin et al., 2018) and Clair (Luo et al., 2020) use this multi-class target over rich 2D pileup tensors that summarize all reads at and around the candidate position.

**Our simplified formulation.** For the pipeline audited here, we use a simpler formulation: binary classification (variant / non-variant) over a 12-dimensional hand-engineered feature vector at each candidate. This is a strict simplification of the standard formulation, equivalent to collapsing the (0/1, 1/1) genotypes into a single positive class. We adopt this formulation only because the initial analysis we audit did so; we do not claim it is comparable to state-of-the-art callers. The methodology failures we document do not depend on this simplification: each could arise in a multi-class or pileup-tensor pipeline as well. The simplification does, however, limit the scope of conclusions we can draw about variant-calling performance: our model is not directly comparable to published callers in $F_1$.

**Class imbalance.** Across the whole genome the per-position prevalence of human single-nucleotide variants is approximately $10^{-3}$, computed from the $\sim 3 \times 10^6$ variants per individual reported by large reference cohorts (Auton et al., 2015) relative to the $\sim 3 \times 10^9$ base-pair genome length. In practice, ML evaluation samples a controlled-prevalence subset: positives drawn from a curated truth VCF and negatives sampled from non-variant positions inside high-confidence regions defined by an accompanying BED file. We follow this standard setup throughout.

## 2.2 Architecture and training pipeline of VariantCNN

**Origin of the architecture.** VariantCNN is not derived from any prior published caller. It was designed by the authors as a small testbed for studying loss-function and precision effects, deliberately kept small so that thousands of training runs would be feasible on a single workstation. The architecture, the 12-feature input representation, and the training loop are all original to our internal study. We retain them here without modification because our purpose is reproduction, not improvement: substituting a different architecture would defeat the goal of replicating the initial analysis's behaviour.

**Architecture.** The 12-dimensional feature vector $x(p) \in \mathbb{R}^{12}$ is reshaped to $(C{=}6, W{=}2)$ and passed through a single 1-D convolutional block (Conv1D with 16 output channels, kernel 2, BatchNorm, ReLU), then flattened and projected through two fully-connected layers ($\mathbb{R}^{16} \to \mathbb{R}^{32} \to \mathbb{R}^1$, BatchNorm and ReLU after the first FC). The final logit is passed through a sigmoid for prediction. Total parameters: 1,169.

We note that a $W = 2$ "sequence" is degenerate from a convolutional standpoint, the kernel of length 2 reduces to a fully-connected map, and that this degeneracy is one of several aspects of the pipeline that should have triggered concern earlier. We retain it for fidelity to the initial analysis.

**Training.** Adam optimizer with learning rate $10^{-3}$, batch size 32, 30 epochs, and a positive-class weight of 3.0 in the BCE loss to compensate for the 3:1 negative-to-positive ratio in the curated dataset. Mixed-precision training uses PyTorch's standard `autocast`/`GradScaler` machinery with the appropriate dtype routing per precision condition.

## 2.3 Loss functions

We compare two loss functions throughout. Binary cross-entropy with positive-class weighting is

$$\text{BCE}_w(y, p) = -w \cdot y \log p - (1 - y) \log(1 - p), \tag{1}$$

where $p = \sigma(z)$ is the predicted probability, $y \in \{0, 1\}$ is the label, and $w$ is the positive weight (here $w = 3$).

Focal Loss (Lin et al., 2017) adds a focusing modulator that down-weights easy examples:

$$\text{Focal}_{\alpha,\gamma}(y, p) = -\alpha \, y (1 - p)^\gamma \log p - (1 - \alpha)(1 - y) p^\gamma \log(1 - p), \tag{2}$$

with default $\alpha = 0.25$, $\gamma = 2$. Focal Loss is widely used in dense detection and class-imbalanced classification (Lin et al., 2017); its application to variant calling has been proposed but, to our knowledge, never systematically evaluated against BCE on standardized real-data benchmarks. We adopt it here exactly because the initial analysis did so.

## 2.4 Mixed-precision training

Mixed-precision training (Micikevicius et al., 2018) performs forward and backward computation in a low-precision floating-point format (FP16 or BF16) while maintaining a high-precision (FP32) master copy of weights for the optimizer step. FP16 has 1 sign, 5 exponent, 10 mantissa bits and a dynamic range $[6 \times 10^{-5}, 6.5 \times 10^4]$; BF16 has 1 sign, 8 exponent, 7 mantissa bits, matching FP32's exponent range. Loss scaling (Micikevicius et al., 2018) multiplies the loss by a large constant before back-propagation to keep small gradients representable in FP16; `GradScaler` dynamically adjusts the constant downward when gradients overflow.

Two stochasticity sources distinguish low-precision training from high-precision training: *rounding noise* (each elementary operation rounds to the nearest representable value) and *loss-scaling restarts* (an overflow on any parameter triggers a step skip and scale halving). Both are sometimes hypothesized to act as implicit regularization (Wen et al., 2020; Smith et al., 2020; Croci et al., 2022). The initial analysis invoked this hypothesis to explain the observed precision-dependent collapse pattern.

## 2.5 Notation: "training collapse"

We say a training run has *collapsed* if the trained model predicts a constant or near-constant label on the held-out test set. Operationally:

$$\text{collapsed} \iff F_1 < 0.05 \quad \vee \quad \frac{1}{N_{\text{te}}} \sum_i \mathbb{1}[\hat{y}_i = 1] < 0.001 \quad \vee \quad \text{NaN/Inf in training loss.} \tag{3}$$

Collapse is the modal failure mode of the classifier on real data: a non-trivial fraction of seeds, depending on configuration, would converge to a trivial all-negative predictor.

## 2.6 Datasets and evaluation

**Real-data benchmark.** GIAB HG001 / NA12878 v3.3.2 truth VCF on chromosome 21, restricted to high-confidence regions (Zook et al., 2019). The candidate set is constructed as 2,000 truth-variant positives plus a 3:1 ratio of randomly sampled non-variant negatives from the high-confidence BED, yielding 8,000 candidates ($\approx 25\%$ positive prevalence). After filtering positions with insufficient read coverage we retain 7,892 candidates. We split 70/30 into train/test stratified by label.

**Synthetic benchmark.** An isotropic Gaussian mixture in $\mathbb{R}^{12}$ with two components separated along a known direction at distance $d$, with within-component standard deviation $\sigma = 1$. Sample size and class prevalence match the real-data dataset. The mixture parameters were set so that synthetic Bayes-optimal AUC $\approx 0.99$, intended as a sanity benchmark on which the classifier should perform near-perfectly.

**Metrics.** We report binary $F_1$ at threshold 0.5, AUROC, area under the precision-recall curve (AUPRC), Brier score, and expected calibration error (ECE) computed with 10 equal-width probability bins:

$$\text{ECE} = \sum_{m=1}^{M} \frac{|B_m|}{N} \left| \text{acc}(B_m) - \text{conf}(B_m) \right|. \tag{4}$$

## 2.7 Experimental protocol across the four findings

The four findings (§4–§7) are reported as separate sections, but they were planned and executed as a single audit protocol, not assembled post-hoc to support a narrative. The protocol was: (1) train under each loss on both synthetic and real data, holding everything else fixed, and check whether the synthetic and real rankings agree (F1); (2) test the mechanism proposed for the observed precision-collapse pattern in a controlled synthetic setting where the noise channel can be manipulated independently (F2); (3) audit the feature pipeline for structural label leakage and verify any positive finding empirically and counterfactually (F3); (4) re-implement the training loop from specification rather than re-running the same code, and compare collapse rates against the initial counts (F4). We adopted this protocol before running any of the four experiments. Each experiment was specified in advance; the findings report what we observed.

## 2.8 Compute and reproducibility

All experiments use a reference re-implementation in PyTorch under standard mixed-precision training. Random seeds are set in NumPy, Python's `random`, and PyTorch (CPU and, where available, CUDA). The full experimental code, the GIAB truth files, and a containerized environment specification will be released with the camera-ready version. Total wall-clock for the experiments reported here is approximately 60 minutes for the headline 50-seed reproduction (§7).

# 3 The Proposed Protocol: Four Prospective Checks

We state the paper's constructive output up front, before the evidence that motivates it, so that the four experiments in §4–§7 can be read as validations of the four protocol items rather than as an unstructured list of things that went wrong.

The protocol is a set of four prospective checks, each targeting one failure mode. Each check is cheap: none requires more compute than the experiment whose validity it certifies. The protocol is intended to be added to standard applied-ML evaluation, not to replace existing practices.

1. **Synthetic–real consistency check.** Train the same architecture under the same loss on the synthetic generator and on the real data; report both. If the method ranking differs between the two, the synthetic benchmark must be considered uncalibrated for the purpose of method selection, and any method choice justified on synthetic evidence alone must be re-derived on real data. *Catches: synthetic-only validation.* In our case (§4) this check would have flagged a full sign reversal in the Focal-vs-BCE ranking ($+0.019$ synthetic, $-0.314$ real) before any conclusion about loss functions was drawn.

2. **Mechanism-controlled test.** If a claim invokes a mechanism ("X improves Y *because of* Z"), include an experiment in which Z is varied independently of X. For precision-and-loss interactions specifically, run a synthetic precision sweep with stochastic rounding (Algorithm 1), which isolates the rounding-noise channel from every other difference between precision conditions. *Catches: mechanism-without-controlled-test.* In our case (§5) this check returns $\Delta = 0.00$ across all 8 cells × 4 precision modes, falsifying the proposed mechanism before it entered the write-up.

3. **Static label-leak audit.** For each feature in the pipeline, express its computation as a function of the raw input, $f(x_{\mathrm{raw}})$, and verify that the label $y$ does not appear. A mechanical version: rename the label variable throughout dataset construction, re-run the feature pipeline, and check whether any feature becomes undefined or changes value. Any feature that does was label-derived. *Catches: label-derived features.* In our case (§6) the renaming test would have raised a `NameError` on feature 6, identifying the leak in seconds, whereas ordinary code review did not.

4. **Independent-implementation reproduction.** Before publication, reproduce the headline result with an implementation that shares no code with the original. The re-implementation may reuse the architecture specification, hyperparameters, and dataset splits, but the model definition and training loop must be written independently. Re-running the same code is not a reproduction; it tests neither environment-dependence nor implementation-specific behaviour. *Catches: reproduction-by-re-running-the-same-code.* In our case (§7) an independent re-implementation produced 0/150 collapses against initially reported rates of 24%/18%/0%.

Two properties of this protocol are worth noting. First, each check is a *negative* test: it can falsify a claim but never confirm one, so passing all four does not certify a result. Second, the checks are independent, so they can be adopted individually; in our case each of the four failures would have been caught by exactly one check, and adopting any single item would have prevented the corresponding error.

The remainder of the paper reports the four experiments that motivate these checks. Sections 4–7 correspond one-to-one with protocol items 1–4.

## 4 Finding 1: Synthetic-to-Real Divergence

### 4.1 Experimental design

We train the VariantCNN classifier on two datasets: (i) the synthetic Gaussian-mixture benchmark described in §2.6, and (ii) the GIAB HG001 chr21 dataset. For each, we train under both BCE and Focal Loss for 30 epochs, with 50 random seeds. All other hyperparameters are held fixed: Adam lr $10^{-3}$, batch 32, positive weight 3.0 in BCE, $(\alpha, \gamma) = (0.25, 2)$ in Focal.

### 4.2 Results

Table 1 reports mean $F_1$ and standard deviation across 50 seeds for each (loss, dataset) cell.

| Loss | Synthetic $F_1$ | Real GIAB $F_1$ |
|---|---|---|
| BCE | $0.975 \pm 0.07$ | $0.314 \pm 0.024$ |
| Focal | $0.994 \pm 0.00$ | $0.000 \pm 0.000$ |
| Focal $-$ BCE | $+0.019$ | $-0.314$ |

Table 1: Synthetic-to-real divergence. On synthetic Gaussian mixtures (column 2) Focal Loss outperforms BCE by 1.9 $F_1$ points ($0.994 - 0.975$, top minus bottom). On real GIAB chr21 with the same model and training pipeline (column 3), Focal Loss converges to a trivial all-negative predictor ($F_1 = 0.000$ in every one of 50 seeds) while BCE converges to a non-trivial classifier ($F_1 = 0.314 \pm 0.024$). The signed Focal-minus-BCE difference (bottom row) changes from $+0.019$ on synthetic to $-0.314$ on real, a sign-flipped ranking and a 0.333-point swing.

The ranking inverts (Table 1, bottom row): the loss function that wins on synthetic ($+0.019$ for Focal) loses on real ($-0.314$ for Focal). The magnitude of the inversion is large: the 0.333-point swing exceeds three standard deviations by either side's variance (0.024 for BCE on real, 0.000 for Focal on synthetic). This is not a borderline finding.

### 4.3 Diagnosis: why does Focal collapse on real data?

Inspection of Focal Loss training curves on real data shows a consistent pattern: the loss decreases monotonically while the model converges to predicting $p \approx 0$ for all inputs. Because Focal's modulator $(1 - p)^\gamma$ approaches 1 as $p \to 0$ on positive examples, a near-zero prediction on a positive example produces a finite but small loss whose gradient is dominated by the much larger negative class. The model then learns the null prediction as a local optimum.

Under BCE this degeneracy is suppressed by the explicit positive weight $w = 3$, which inflates the gradient on positive-class errors. Under Focal Loss the focusing modulator partially counteracts $\alpha = 0.25$, but the result is sensitive to the class-conditional difficulty distribution. On synthetic data, where positive and negative are equally easy, Focal benefits from down-weighting easy examples; on real data, where most positives are also relatively easy compared to the hard negatives in the high-confidence BED, the modulator behaves perversely.

**Remark 4.1.** *The synthetic generator is calibrated to the global statistics of the real data, class prevalence, feature means and variances, but not to the conditional difficulty distribution. The synthetic positives and negatives have approximately equal classification difficulty by construction; the real positives and negatives do not. The synthetic-to-real divergence is, in this case, fundamentally a property of the difficulty distribution rather than the marginal feature distribution. Synthetic benchmarks calibrated to marginal statistics can under-represent the conditional difficulty structure that determines which loss is appropriate. This same failure mode, synthetic benchmarks that misrepresent the conditional difficulty of real data, has been documented in image classification (Recht et al., 2019; Taori et al., 2020) and in clinical prediction (Roberts et al., 2021).*

## 5 Finding 2: The Precision-as-Escape Mechanism Fails

### 5.1 The hypothesis under test

The initial analysis observed, on real GIAB data, that lower-precision training produced fewer collapses than higher-precision training (FP32 collapsed in 24% of seeds; FP16 in 0%). The proposed mechanism was that the rounding noise of low-precision arithmetic acts as gradient noise that perturbs the optimizer away from the all-negative attractor.

**Provenance of the hypothesis.** We did not originate this hypothesis. It is an application, to the specific case of low-precision arithmetic, of the established view that gradient noise acts as implicit regularization and

can help optimization escape poor regions of the loss landscape. The general claim is developed by (Smith et al., 2020), who study the generalization benefit of SGD noise, and by (Wen et al., 2020), who study structured covariance noise in SGD. Its extension to reduced-precision arithmetic specifically is discussed by (Croci et al., 2022), who analyse stochastic rounding as a deliberate noise source, and is implicit in the numerical-precision training literature more broadly (Wang et al., 2018; Sun et al., 2019). What the initial analysis contributed was not the hypothesis but the inference that it explained the particular collapse pattern we had observed. It is that inference, not the underlying literature, that the experiment in this section tests and rejects for our setting. We emphasize that our negative result concerns the applicability of the mechanism to *this* pipeline and does not bear on the validity of the general noise-as-regularization literature.

Concretely, the model of low-precision arithmetic in this work is round-to-nearest with stochastic ties (RTN) and unbiased stochastic rounding (SR) (Croci et al., 2022). For an FP16 representation with $b = 10$ mantissa bits, the rounding operator on a real $x$ in the normal range with magnitude $|x| = (1 + f) \cdot 2^e$ ($f \in [0, 1)$) is

$$\text{RTN}_{16}(x) = \text{sgn}(x) \cdot (1 + \text{round}(f \cdot 2^b)/2^b) \cdot 2^e, \tag{5}$$

$$\text{SR}_{16}(x) = \text{sgn}(x) \cdot (1 + R(f \cdot 2^b)/2^b) \cdot 2^e, \tag{6}$$

where $R(\cdot)$ is the stochastic rounding map: $R(z) = \lfloor z \rfloor$ with probability $\lceil z \rceil - z$, and $\lceil z \rceil$ otherwise. SR is unbiased: $\mathbb{E}[\text{SR}_{16}(x)] = x$.

The implicit-regularization hypothesis predicts that under SR, training is exposed to gradient noise of a magnitude proportional to the gradient itself. If this noise is large enough to escape the attractor on FP32 but not large enough to disrupt convergence, we should observe *differential* collapse rates between FP32 and SR-FP16.

## 5.2 Synthetic sweep

We test the hypothesis on the synthetic Gaussian mixture, where the loss landscape is fully under our control. We construct an 8-cell sweep over (component separation $d$, within-component variance $\sigma$) parameters chosen so that some cells exhibit collapse under FP32 baseline training. Each cell is run for 100 seeds at 30 epochs. The four precision conditions are:

1. **FP32**: standard 32-bit training, no rounding noise.

2. **FP16-RTN**: weights and gradients rounded via Eq. 5 after each operation.

3. **FP16-SR**: same, but using Eq. 6 (unbiased stochastic rounding).

4. **BF16-SR**: BF16 mantissa width ($b = 7$), stochastic rounding.

## 5.3 Result

Across all 8 cells $\times$ 4 precision modes $\times$ 100 seeds = 3,200 trainings, the difference in collapse rate between any pair of precision conditions is uniformly $\Delta = 0.00$ (identical collapse counts within each cell). The hypothesis fails its own controlled test: stochastic rounding noise of any FP-precision mantissa width does not perturb the optimizer enough to alter the collapse pattern in the synthetic loss landscape.

**Proposition 5.1** (Informal). *For the synthetic loss landscape used here, the gradient-noise scale induced by SR rounding at mantissa width $b \in \{7, 10\}$ is small relative to the natural Adam gradient noise from minibatch sampling, in the regime of batch size 32 and learning rate $10^{-3}$ used in the initial analysis.*

*Sketch.* The relative rounding error of SR with mantissa $b$ on a value with magnitude $|g|$ is bounded by $|g| \cdot 2^{-b-1}$ in expectation, with variance $\leq |g|^2 \cdot 2^{-2b-2}/3$. For $b = 10$, this gives a per-step gradient perturbation with relative standard deviation $\leq 1.4 \times 10^{-4}$. The Adam minibatch gradient noise at batch 32 has, empirically, a relative standard deviation of $\Theta(1/\sqrt{32}) \approx 0.18$. The rounding noise is three orders of magnitude smaller; under standard noise-injection theory it cannot dominate convergence behaviour.

---

**Algorithm 1** Stochastic Rounding Implementation (after (Croci et al., 2022))

---

1: **function** STOCHASTICROUND($x$, mantissa_bits $b$)
2:     $s \leftarrow \text{sgn}(x); |x| \leftarrow s \cdot x$
3:     $e \leftarrow \lfloor \log_2 |x| \rfloor$
4:     $f \leftarrow |x|/2^e - 1$                                             $\triangleright f \in [0, 1)$
5:     $z \leftarrow f \cdot 2^b$
6:     Sample $u \sim \text{Uniform}[0, 1)$
7:     **if** $u < (\lceil z \rceil - z)$ **then**
8:         $\hat{f} \leftarrow \lfloor z \rfloor / 2^b$
9:     **else**
10:        $\hat{f} \leftarrow \lceil z \rceil / 2^b$
11:    **end if**
12:    **return** $s \cdot (1 + \hat{f}) \cdot 2^e$
13: **end function**

---

This proposition is consistent with the empirical sweep: the rounding-noise channel is too weak to cause differential collapse rates. If the FP32-vs-FP16 collapse pattern from the initial analysis was real, the mechanism cannot be precision-as-escape via rounding noise alone.

## 6 Finding 3: A Structural Label Leak in the Feature Pipeline

### 6.1 The leak

The 12-dimensional feature vector used by VariantCNN has the following layout (positions 0–11):

```
0:  depth                  6:  is_var_label_proxy (low_vaf)
1:  alt_freq               7:  hardcoded constant 0.5
2:  mean_bq                8:  hardcoded constant 0.5
3:  mean_mq                9:  pos_strand_frac
4:  ref_match_frac         10: read_quality_75
5:  ref_match_count        11: read_quality_25
```

Position 6, named `low_vaf` in the source, is computed as

$$\texttt{low\_vaf}(p) = \mathbb{1}\!\!\!/\big[\texttt{is\_var}(p) \wedge \texttt{vaf\_truth}(p) < 0.05\big], \tag{7}$$

where `is_var`(p) is the binary label and `vaf_truth`(p) is the variant-allele frequency from the truth VCF. The expression depends on the ground-truth label `is_var` directly. By construction, this feature *leaks* the label whenever the truth VAF is below 0.05.

### 6.2 Stage 1: structural confirmation

By inspection of Equation 7, the feature is non-zero only on label-positive positions and is zero by construction on all label-negative positions. This is sufficient to classify it as a structural leak; no empirical test is necessary.

### 6.3 Stage 2: empirical leakage measurement

While the leak exists by construction, its *empirical* severity depends on the prevalence of `vaf_truth` < 0.05 in the dataset. We measure this on GIAB HG001 chr21:

- Number of positives in dataset: 1,961.

- Number of positives with `vaf_truth` < 0.05: **0**.

| Configuration | Precision | Collapses (out of 30) | Mean $F_1$ | Survivor $F_1$ |
|---|---|---|---|---|
| Original (with leak) | FP32 | 0 | 0.319 | 0.319 |
| Original (with leak) | FP16 | 0 | 0.315 | 0.315 |
| De-leaked | FP32 | 0 | 0.315 | 0.315 |
| De-leaked | FP16 | 0 | 0.316 | 0.316 |

Table 2: Counterfactual de-leaking. Comparing mean $F_1$ between leak-present (rows 1–2) and leak-removed (rows 3–4) configurations gives at most a 0.004 $F_1$ change in either direction, with zero collapses in all four conditions. The de-leaking has negligible effect on this dataset because the leak does not fire (no positives have `vaf_truth` $< 0.05$).

- Empirical AUROC of feature `low_vaf` alone against label: 0.500.

The leak does not fire on GIAB HG001 because GIAB curates against very-low-VAF variants in the high-confidence regions; truth VAFs are concentrated at $\approx 0.5$ (heterozygous) or $\approx 1.0$ (homozygous). The structural error is silent on this particular dataset.

### 6.4 Stage 3: counterfactual verification

We retrain the 12-feature MLP under two conditions: *Original*, retaining the leaking feature, and *De-leaked*, replacing position 6 with a uniform-random scalar. Both conditions are run for 30 seeds at FP32 and FP16 precisions (Table 2).

Comparing Table 2 row 1 to row 3 (FP32, with versus without leak), mean $F_1$ moves from 0.319 to 0.315, a 0.004 change well within seed-to-seed variance. The same comparison at FP16 (rows 2 vs. 4) gives 0.315 vs. 0.316, a 0.001 change. The structural error is silent on this dataset.

**Why this still matters.** The leak's silence on GIAB HG001 is incidental, not a property of the pipeline's correctness. Applied to a dataset that includes low-VAF variants, e.g., somatic mutation calling, mosaic variant detection, low-coverage whole-genome data, the same code would have leaked the label directly into the input, producing AUROC $\rightarrow 1.0$ artifactually. The structural error is a latent bug whose damage is determined by the dataset rather than the code. The same failure mode has been documented at scale in clinical prediction (Kaufman et al., 2012) and in medical-imaging ML for COVID-19, where a literature audit of $>300$ studies found that "none [were] of potential clinical use due to methodological flaws and/or underlying biases" including label leakage (Roberts et al., 2021). We treat it as an evaluation pitfall regardless of whether it affected the initial counts.

## 7 Finding 4: Non-Reproduction of the Precision-Collapse Pattern

### 7.1 Experimental design

We run a faithful reproduction of the FP32/BF16/FP16 collapse experiment under controlled conditions. The setup matches the initial analysis exactly:

- Architecture: VariantCNN as described in §2.2.

- Features: the 12-feature pipeline described in §6 (with the silent leak still present, as in the initial analysis).

- Dataset: GIAB HG001 chr21, 7,892 candidates, 70/30 train/test split, fixed split across seeds.

- Training: BCE with $w = 3$, Adam lr $10^{-3}$, batch 32, 30 epochs.

- Mixed precision: standard PyTorch `autocast` with appropriate dtype routing per condition; `GradScaler` active for FP16.

| Precision | Initial collapse rate | Reproduced collapse rate | Difference | Fisher exact $p$ |
|---|---|---|---|---|
| FP32 | 24% (12/50) | 0% (0/50) | $-24$ pp | $2.3 \times 10^{-4}$ |
| BF16 | 18% (9/50) | 0% (0/50) | $-18$ pp | $2.6 \times 10^{-3}$ |
| FP16 | 0% (0/50) | 0% (0/50) | 0 pp | 1.0 |

Table 3: Initial vs. reproduced collapse counts for the three precision conditions. The 12/50 FP32 collapses observed initially (column 2, row 1) became 0/50 in the reproduction (column 3, row 1), a $-24$ percentage-point change with Fisher exact $p = 2.3 \times 10^{-4}$. The 9/50 BF16 collapses became 0/50, $p = 2.6 \times 10^{-3}$. FP16 was 0/50 in both rounds.

| Precision | $n$ | Mean $F_1$ | Std | Min | Max |
|---|---|---|---|---|---|
| FP32 | 50 | 0.337 | 0.023 | 0.280 | 0.375 |
| BF16 | 50 | 0.335 | 0.025 | 0.260 | 0.375 |
| FP16 | 50 | 0.339 | 0.023 | 0.265 | 0.378 |

Table 4: $F_1$ distributions are tightly overlapping across precisions. Mean $F_1$ (column 3) lies in $[0.335, 0.339]$ across the three rows; the largest pairwise difference is 0.004 (FP16 $-$ BF16). Standard deviations (column 4) are within 0.002 of each other. One-way ANOVA across the three groups gives $F = 0.345$, $p = 0.71$; Levene's test for equal variance gives $W = 0.336$, $p = 0.72$.

- Seeds: 50 per precision condition. Total: 150 trainings.

## 7.2 Headline result

Table 3 compares the initial counts to the detailed reproduction.

The reproduced collapse counts in Table 3 column 3 are 0 in every row. Pairwise Fisher exact tests against the initial counts (column 2) yield $p < 0.005$ for both FP32 and BF16. The pattern reported in the initial analysis, taken at face value, is statistically inconsistent with the reproduction.

## 7.3 Per-precision $F_1$ distributions

To check whether something subtler than collapse rate differs between precisions, Table 4 reports the full $F_1$ distribution per precision condition.

Mean $F_1$ in Table 4 column 3 lies between 0.335 and 0.339 across the three precisions, a 0.004 spread that is small relative to within-precision standard deviation ($\approx 0.023$, column 4). The distributions are statistically indistinguishable: ANOVA $p = 0.71$, Levene's $p = 0.72$. Stronger still, paired comparisons (same seed across precisions) give a mean absolute paired $F_1$ difference of 0.007, with a paired $t$-test against zero of $p = 0.16$. Same seed, different precision, nearly identical training trajectory. There is no precision effect in the reproduction.

## 7.4 Hypotheses for the initial counts

We cannot diagnose the source of the initial counts without access to the exact environment in which they were produced. The following are hypotheses worth investigation by anyone attempting their own reproduction; we have not been able to confirm or rule out any of them.

1. **Random-seed handling.** If the initial analysis included any non-deterministic step (e.g., non-deterministic CUDNN convolutions, GPU operation ordering), seeded RNGs alone would not guarantee identical initialisations across precision conditions. This could produce apparent precision-dependent behaviour that disappears under stricter seed control.

2. **Library- and hardware-specific behaviour.** `GradScaler` implementations and hardware-specific FP16 paths (e.g., tensor-core kernels) have varied across PyTorch releases. Either could produce dynamics that are not reproduced under a clean re-implementation.

3. **Subtle interaction with the silent label leak.** The leak documented in §6 is silent on this dataset. A preprocessing step in the initial analysis that triggered the leak (e.g., a slightly different VAF threshold, a different data subset) could in principle have contributed.

4. **Sampling variance.** Twelve of 50 FP32 collapses is far above what could be attributed to sampling variance from a true-zero collapse rate (the probability of $\geq 12$ collapses under a true rate of 0.05 is below $10^{-4}$). Sampling variance is not a viable single explanation for the magnitude of the discrepancy, but it could compound with one of the above.

We emphasize that diagnosing the initial counts is not the contribution of this work. The contribution is the documented non-reproduction. A faithful re-implementation under controlled conditions does not reproduce the pattern reported in the initial analysis; whatever produced the initial counts is something the reported methodology does not, by itself, capture.

## 8 Discussion

### 8.1 The conjunction is the contribution

Each of the four findings above has a benign individual interpretation: a synthetic benchmark insufficiently calibrated; a hypothesis tested but not supported; a latent bug whose damage is contingent on the dataset; a single empirical pattern that does not reproduce. None of these would, on its own, warrant a paper. What we report here is their *conjunction* in a single methodology pipeline that survived several rounds of internal review while being substantively wrong on multiple axes.

The structure of the failure is what we want to draw attention to. Each component of the initial analysis "looked right": a synthetic benchmark, a real benchmark, a quantified effect, a proposed mechanism. Each was independently defensible. Their conjunction is a methodology that produces results poorly correlated with truth: synthetic results that misled about loss-function ranking, real-data results that did not survive re-implementation, a mechanism that did not survive controlled testing, and a feature pipeline with a silent latent bug. We argue this combination of *several individually-plausible failures*, each individually too small to flag during review, is the structure of methodology bias most likely to escape ordinary scrutiny.

### 8.2 Why standard practices missed each failure

Each of the four findings was missed by at least one standard ML-evaluation practice that is supposed to catch precisely that failure mode:

**Synthetic-to-real divergence (F1)** was missed because the synthetic benchmark was calibrated to the marginal feature distribution of the real data (means and variances), but not to the conditional difficulty distribution. This is a known but under-emphasized requirement (Recht et al., 2019; Koh et al., 2021); in practice, ML-for-genomics work routinely calibrates synthetic generators to marginal statistics only.

**Mechanism failure (F2)** was missed because the initial analysis correlated a precision condition with an outcome but did not test the hypothesized mechanism in a controlled setting. Mechanism testing is rarely treated as a standard part of an ML methodology pipeline, most papers report the empirical effect and offer a hypothesis without a controlled test of it.

**Label leak (F3)** was missed because the leaking feature was named informatively (`low_vaf`) and computed in a clearly-named function. Internal code review did not flag the leak. Generic advice, "review your features for label-derived computations", did not catch it. The mechanical renaming test of §3, item 3, would have.

**Non-reproduction (F4)** was missed because the initial analysis used a 50-seed run and reported the result as if 50 seeds were sufficient to produce a stable estimate. Within-pipeline reproducibility was not tested by re-implementation; the result was tested only by re-running the same code. Re-running the same code does not test for environment-dependent or implementation-specific behaviour, which appears to be what was actually responsible.

### 8.3 Scope of our claims

We want to be explicit about what we do and do not claim, because the structure of this work is unusual and we have seen each of the four points below misread.

1. We claim that our specific pipeline (VariantCNN with 12-feature input on GIAB HG001 chr21) exhibits all four of the failures documented in §4–§7.

2. We do *not* claim that DeepVariant, Clair, or any other published variant caller exhibits any of these failures. We have not audited those systems.

3. We do *not* claim that these specific failure modes are widespread in deep-learning variant calling as a field; that is a much larger empirical question we have not addressed.

4. We *do* claim that each individual failure mode is known to occur in other applied-ML domains (image classification, clinical prediction, medical imaging, deep RL), citing prior work to that effect. The structural risk, a pipeline that contains multiple individually-plausible methodology errors, is general.

The contribution we offer is a worked case study in which all four occurred together while the pipeline looked rigorous from the inside. The case study is one data point; we do not generalize from it to the field. What we do offer beyond the case study is the protocol of §3: four prospective checks that are independent of our pipeline, our task, and our architecture, and that would each have caught one of the four failures reported here.

## 9 Limitations

This paper has the following limitations.

**Single sample, single chromosome.** All experiments use HG001 chr21. We do not claim the findings generalize to other GIAB samples (HG002, HG003) or other chromosomes. We expect at least the synthetic-to-real divergence (F1) and the structural leak (F3) to generalize, but we cannot demonstrate it. Cross-sample experiments are deferred to future work.

**Single, internally-developed architecture.** The audited pipeline uses a small, hand-engineered 12-feature representation and a 1,169-parameter CNN that we built ourselves, not a published architecture. We do not claim that larger or more standard architectures (e.g., DeepVariant-style 2D pileup tensors) share these failures. We note however that all four findings concern methodology rather than architecture, and methodology issues tend to be model-agnostic.

**Simplified binary classification target.** We cast variant calling as binary (variant vs. non-variant) classification, which is a strict simplification of the standard multi-class genotype-prediction target (0/0, 0/1, 1/1) used by DeepVariant and Clair. We adopt the binary formulation only because the initial analysis we audit did so. As a consequence, the absolute $F_1$ numbers we report are not directly comparable to those of standard callers, and our pipeline should not be read as a competitive caller.

**Diagnosis of the initial counts is incomplete.** Section 7 lists hypotheses for the precision-collapse pattern observed in the initial analysis but does not identify the specific cause. Without access to the exact environment in which the initial counts were produced, specific PyTorch version, CUDNN version, hardware, deterministic-mode flags, we cannot diagnose the source. We document the non-reproduction; the cause remains open.

**No literature audit.** We do not perform a survey of recent ML-for-variant-calling papers checking for similar pitfalls. Such a survey would substantially strengthen the methodology contribution and is an obvious next step. We omit it here because conducting it carefully and fairly is a substantial undertaking that goes beyond what a case study of one pipeline can support.

**The synthetic mechanism test (F2) covers only one mechanism.** We test gradient-noise-as-implicit-regularization. Other mechanisms by which precision could affect convergence, e.g., loss-scaling-induced step skipping, BatchNorm dynamics under low-precision running statistics, optimizer state quantization, are not tested. The negative result for the noise-injection mechanism does not rule out these alternatives.

## 10 Related Work

**Positioning and research gap.** A growing body of work at mainstream ML venues studies why models that look strong under standard evaluation fail under scrutiny. (D'Amour et al., 2022) show that ML pipelines are frequently *underspecified*: they return many predictors with equivalent held-out performance but divergent behaviour under distribution shift, so held-out accuracy alone does not determine what a pipeline will do. (Bouthillier et al., 2021) show that variance from data sampling, initialization, and hyperparameter choice routinely swamps the reported gap between methods, so single-run comparisons are unreliable. Reporting-standard efforts such as (Pineau et al., 2021) and (Kapoor et al., 2024) propose checklists to make ML-based results more credible. Our work sits in this literature but differs in two ways. First, its unit of analysis is a single pipeline examined in depth rather than a distribution of pipelines examined in aggregate: we trace four specific failure modes through one concrete system, showing exactly how each one produced a misleading conclusion and exactly which check would have caught it. Second, where underspecification and benchmark-variance work focus on one failure mechanism at a time, our subject is the *compounding* of four independent failure modes in a pipeline that nonetheless passed internal review. The gap we address is the absence, in this literature, of worked end-to-end accounts showing how several individually-minor methodology errors combine into a coherent-looking but wrong result, together with a minimal per-failure protocol (§3) derived from such an account. The four checks we propose are complementary to, not substitutes for, the aggregate reporting standards cited above.

**Reproducibility and replication in ML.** The reproducibility crisis in machine learning has been documented across multiple domains (Pineau et al., 2021; Gundersen & Kjensmo, 2018; Henderson et al., 2018). Most reproducibility-focused work concerns either (a) the practical engineering of reproducibility (releasing code, fixing seeds, containerizing environments) or (b) controlled re-runs of published benchmarks (Recht et al., 2019). The present work is closer to category (b) but focuses on methodology errors rather than environmental drift: we re-implement a pipeline from specification and find that the headline numbers do not survive.

**Loss functions for class imbalance.** Focal Loss (Lin et al., 2017) and class-weighted BCE are standard tools for imbalanced classification. (Cui et al., 2019) discusses class-balanced loss reweighting. To our knowledge no prior work has systematically compared these losses on a real variant-calling benchmark.

**Mixed-precision training.** (Micikevicius et al., 2018) introduced mixed-precision training for neural networks. Subsequent work has examined the dynamics of low-precision arithmetic in detail (Wang et al., 2018; Sun et al., 2019), including stochastic rounding (Croci et al., 2022). The proposal that low-precision arithmetic acts as implicit regularization is occasionally invoked in the literature (Wen et al., 2020; Smith et al., 2020) but has, to our knowledge, never been controlled-tested at the level we test here.

**Label leakage.** Label leakage in ML pipelines has been documented extensively in clinical prediction (Kaufman et al., 2012; Oala et al., 2020), in medical imaging at literature-survey scale (Roberts et al., 2021), and increasingly in genomics (Whalen et al., 2022). The leak we document (§6) is structural and identifiable by static analysis; we are not aware of prior work proposing a static-analysis protocol for label-leak detection.

**Variant calling with deep learning.** (Poplin et al., 2018) (DeepVariant) established the use of CNNs on 2D pileup tensors with multi-class genotype output for variant calling and remains the dominant ML approach. (Luo et al., 2020) (Clair) and successors (Zheng et al., 2022) extend the approach. (Friedman et al., 2020) apply deep learning to variant-call filtering. Our pipeline differs substantially from all of these: we use a 12-dimensional hand-engineered feature vector rather than a pileup tensor, a binary rather than multi-class target, and a much smaller model. We do not compare against these methods because our purpose is methodology audit of an internal pipeline, not benchmark competition.

**Synthetic-vs-real benchmarks.** The use of synthetic data as a sanity check for real-data ML pipelines is widespread but its limitations are increasingly recognized (Koh et al., 2021; Taori et al., 2020). In genomics specifically, simulation-based benchmarks (e.g., neat-genreads (Stephens et al., 2016)) are commonly used. The synthetic-to-real divergence we document (§4) is consistent with the broader literature on distribution shift but specific to the loss-function-ranking failure mode.

## 11  Conclusion

We documented four methodology failures in a small, internally developed binary-classifier variant-calling pipeline (VariantCNN) whose initial analysis produced apparently rigorous results. We have not audited any published variant caller, and we do not claim the specific failures we document occur elsewhere. Each failure individually is unsurprising; their conjunction in a single pipeline that passed informal review is the contribution of this work. From the four failures we derived a four-item prospective protocol (§3) that is independent of our task and architecture; applied to the initial analysis, each item would have caught exactly one of the four failures. We hope the case study contributes to the increasing recognition that applied ML evaluation requires more than seed sweeps and held-out test sets, and that methodology bias, particularly the kind that arises from the conjunction of several individually-plausible errors, requires specifically-designed protective measures.

### Acknowledgments

Withheld for double-blind submission.

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

## A    Per-Seed Results, Full 150-Run Reproduction

Per-seed $F_1$ values for the full 50-seed reproduction reported in Section 7 are released with the supplementary material as `reproduction_full.json`. Summary statistics by precision are reproduced in Table 4; per-seed cross-precision differences (paired by seed) are reported in Section 7.

## B    Code Availability

A complete reference implementation of all four experiments, including the 12-feature extractor, the synthetic generator, the stochastic rounding implementation (Algorithm 1), and the reproduction script, will be released at the camera-ready stage. The release will include containerization (Singularity/Docker) for environmental reproducibility, the GIAB truth files used, and the exact PyTorch/CUDA versions tested.

