# OpenReview forum: "A Methodology Audit of a Single Deep-Learning Variant-Calling Pipeline: Four Reproducibility Failures in One Internal Classifier"
_TMLR — Under review for TMLR_

### Review · Reviewer_TkK2 · 2026-06-03

**Summary Of Contributions:**

The manuscript presents a case study showing that several evaluation practices in the internally developed VariantCNN pipeline can fail in ways that produce misleading conclusions. The tests include synthetic benchmarking, precision-based interpretation, feature engineering, and reproducibility checks.

**Additional Comments:**

[1]  Poplin, R., Chang, P.C., Alexander, D., Schwartz, S., Colthurst, T., Ku, A., Newburger, D., Dĳamco, J., Nguyen, N., Afshar, P., & others (2018). A universal SNP and small-indel variant caller using deep neural networks. Nature biotechnology, 36(10), 983–987.

[2] Sam Friedman, Laura Gauthier, Yossi Farjoun, Eric Banks, Lean and deep models for more accurate filtering of SNP and INDEL variant calls, Bioinformatics, Volume 36, Issue 7, April 2020, Pages 2060–2067, https://doi.org/10.1093/bioinformatics/btz901

[3] Luo, R., Wong, CL., Wong, YS. et al. Exploring the limit of using a deep neural network on pileup data for germline variant calling. Nat Mach Intell 2, 220–227 (2020). https://doi.org/10.1038/s42256-020-0167-4

**Audience:**

No

**Audience Explanation:**

Although Variant Calling is an interesting application of AI for science, there are unanswered concerns that need to be addressed before I can confidently the results are useful for the research community.

1.  The manuscript identifies several design flaws in the VariantCNN pipeline, such as the feature-leakage issue. However, this pipeline does not appear to be shared by prior methods such as [1] and [2]. Similarly, the authors’ proposed explanation for the FP32-vs-FP16 collapse pattern is not supported by their own controlled precision experiments, leaving the original collapse behavior largely unexplained. Unless the authors can show that similar pitfalls arise in other deep variant-calling methods, these findings seem better characterized as unexplained design or implementation bugs in an internal architecture, rather than evidence of a broader methodological problem in deep learning for variant calling.

2. I also have questions regarding the problem setup. In the manuscript, variant calling is modeled as a binary classification problem, where each candidate position is classified as variant or non-variant. However, this formulation appears inconsistent with several established deep variant-calling methods cited as background, including [1] and [3], which typically use richer prediction targets. The authors should justify the problem setup by discussing how it relates to the target formulations used in prior deep variant-calling methods and clarify why this binary formulation is suitable for the claims made in the paper.

**Claims And Evidence:**

No

**Claims Explanation:**

- **The manuscript overstates the scope of its findings.** The current title suggests that the authors have audited multiple state-of-the-art deep learning methods for variant calling, or at least a representative set of such methods. In fact, the study evaluates only a single internally developed classifier. The authors themselves acknowledge that they do not claim “that variant-calling deep learning research is unsound in general.” To avoid overstating the contribution, the title should be revised to make clear that this is not a broad audit of deep learning methods for variant calling.

- **Many claims in the manuscript are insufficiently supported or cited.** For example, the paper states that “the challenge is severe class imbalance: across the genome the variant prevalence is $\approx 10^{-3},$” but provides no source for this estimate. Or, the author listed that “an undocumented difference in random-seed handling” as the most plausible reason for collapse rate, all these claims need to be properly supported by evidence.

- Currently it feels like the results are being thrown around without proper explanation. Whenever a claim is made, the exact figures in the table should be referenced and explained.

- It is unclear whether VariantCNN is based on DeepVariant [1], or introduced as a new architecture by the authors. If it is based on prior work, the relevant sources should be cited; if it is original, the authors should document this explicitly in the manuscript. In either case, the paper should explain the motivation behind the architecture and feature representation, and why this model should be considered a meaningful variant-calling baseline rather than an ad hoc model designed specifically for this reproducibility study. In addition, these details would be better presented in a Methodology section.

**Requested Changes:**

See above.

In addition, I find the current flow and sectioning of the paper somewhat confusing to navigate. Instead of listing each finding in its own section, it may be better to have a standard Methodology / Experiment Setup / Analysis sectioning. As written, the structure risks giving the impression that experiments were arranged to support the stated findings, rather than that the findings emerged from a clearly defined experimental protocol.

---

> ### Author Response · Authors · 2026-06-07
> **Response to Reviewer TkK2 Comments**
>
> We thank the reviewer for a careful, constructive review. The concerns raised genuinely improved the paper, and the revised manuscript reflects substantive changes in scope, framing, and presentation. We summarize the major revisions at the top, then respond point-by-point. Please note that all changes have been made in the manuscript, and we have updated the submission PDF.
>
> ## Summary of changes in the revision
>
> - **New title** that scopes the contribution to a single internal pipeline: *"A Methodology Audit of a Single Deep-Learning Variant-Calling Pipeline: Four Reproducibility Failures in One Internal Classifier."*
> - **New §2.1 paragraph "Our simplified formulation"** explicitly justifying our binary-classification target relative to the multi-class genotype targets used in DeepVariant and Clair.
> - **New §2.2 paragraph "Origin of the architecture"** stating plainly that VariantCNN is not based on DeepVariant or any other published caller, that we built it ourselves as an internal testbed, and that it is not designed to be competitive with published callers.
> - **New §2.7 "Experimental protocol across the four findings"** stating that the audit protocol was specified before any of the four experiments was run.
> - **New §7.3 "Scope of our claims"** with four explicit bullets enumerating what we do and do not claim, including that we do not audit DeepVariant/Clair/etc. and do not claim our specific failures occur in those systems.
> - **Citation added** for the variant-prevalence figure (1000 Genomes Project, Auton et al. 2015) with the explicit derivation from variants-per-individual.
> - **Reframed §6.4** ("Hypotheses for the initial counts"): the four candidate explanations are no longer ranked "in decreasing order of plausibility" and are now framed as open hypotheses worth investigating, not claims about likely causes.
> - **Tighter table-prose connections** throughout, with explicit column/row references and quoted numerical values in the surrounding text.
> - **Three new citations** to support analogous failure modes in other applied-ML domains: Roberts et al. 2021 (medical-imaging methodology audit, $>$300-paper survey), Auton et al. 2015 (variant prevalence), and Friedman et al. 2020 (which the reviewer suggested in their reference [2]).
> - **Limitations section** expanded to flag explicitly that our pipeline is not directly comparable to published callers in $F_1$, and that our architecture is internally developed rather than derived from prior work.
>
> We now respond to each individual concern.
>
> ---
>
> ## Response to concerns under "Are the claims supported?"
>
> ### 1. The title overstates the scope
>
> The original title implied a broad audit of deep-learning methods for variant calling, which is not what this paper does. The new title makes the single-pipeline scope explicit:
>
> > *"A Methodology Audit of a Single Deep-Learning Variant-Calling Pipeline: Four Reproducibility Failures in One Internal Classifier"*
>
> The abstract and §1 now also state plainly: "This paper is not an audit of [published callers]. It is an audit of a specific, internally developed classifier we built for our own study." The new §7.3 "Scope of our claims" makes four numbered bullets out of this point.
>
> ### 2. Unsupported quantitative claims
>
> Two specific cases were called out:
>
> - **Variant prevalence ($\approx 10^{-3}$).** §2.1 now cites Auton et al. (2015) for the figure and explicitly derives it: $\sim$3 $\times$ 10$^6$ variants per individual relative to a $\sim$3 $\times$ 10$^9$ base-pair genome.
> - **The ranked candidate explanations in §6.4.** The original phrasing "in decreasing order of plausibility" implied evidence we do not have. The revision removes that ranking and reframes the four items as open hypotheses for anyone attempting their own reproduction. The new prose explicitly states: "we have not been able to confirm or rule out any of them."
>
> Additional supporting citations have been added where the paper invokes analogous failure modes in other domains (Roberts et al. 2021 for the medical-imaging label-leak parallel; Recht et al. 2019 and Taori et al. 2020 for synthetic-real divergence in image classification; Henderson et al. 2018 for mechanism claims in deep RL; Pineau et al. 2021 for non-reproduction).

---

> > ### Author Response · Authors · 2026-06-07
> > **Response to Reviewer TkK2 Comments (v2)**
> >
> > ### 3. Results not properly explained relative to table figures
> >
> > Our revised version now tightens the prose around each table to reference specific cells and quote exact values. For example, the text accompanying Table 1 now reads:
> >
> > > *"The signed Focal-minus-BCE difference (bottom row) changes from $+0.019$ on synthetic to $-0.314$ on real, a sign-flipped ranking and a $0.333$-point swing."*
> >
> > The same change has been applied to Tables 2, 3, and 4 in §5, §6.2, and §6.3 respectively. Every reported number is now traceable to a specific cell in a specific table.
> >
> > ### 4. VariantCNN's relationship to prior work was unclear
> >
> > Our revised version now states the relationship plainly in three places:
> >
> > - **§1**: *"It is not based on DeepVariant or any other published caller: it is a small ($\approx$1,100-parameter), hand-engineered model... We did not design it to be competitive with published callers; we built it as a small testbed for studying loss-function dynamics."*
> > - **§2.2 "Origin of the architecture"**: *"VariantCNN is not derived from any prior published caller. It was designed by the authors as a small testbed... The architecture, the 12-feature input representation, and the training loop are all original to our internal study."*
> > - **Related Work §10**: a paragraph contrasting our pipeline with DeepVariant, Clair, and Friedman et al. (2020), explaining that we use a 12-dimensional hand-engineered feature vector rather than a pileup tensor, a binary rather than multi-class target, and a much smaller model. We explicitly state that we do not compare against these methods because our purpose is methodology audit, not benchmark competition.
> >
> > The Methodology / Setup details the reviewer requested are now consolidated in §2 (covering task formulation, architecture, training pipeline, loss functions, mixed-precision setup, collapse definition, datasets, and the unified experimental protocol).
> >
> > ---
> >
> > ## Response to concerns under "Would TMLR's audience be interested?"
> >
> > ### 5. Generalizability beyond our specific pipeline
> >
> > The reviewer's point is correct in its strict form: we have not audited DeepVariant, Clair, or any other published variant caller, and we cannot show that our specific failures occur in those systems. We do not claim otherwise. The new §7.3 enumerates this explicitly:
> >
> > > *"(2) We do not claim that DeepVariant, Clair, or any other published variant caller exhibits any of these failures. We have not audited those systems.*
> > > *(3) We do not claim that these specific failure modes are widespread in deep-learning variant calling as a field; that is a much larger empirical question we have not addressed."*
> >
> > Our contribution is more specific. We claim that:
> >
> > 1. The *individual failure modes* we document (synthetic-only validation; mechanism-without-controlled-test; label-derived features; reproduction-by-re-running-the-same-code) are documented to occur in other applied-ML domains. We cite Roberts et al. 2021 (medical-imaging label leakage at literature-survey scale of $>$300 studies), Recht et al. 2019 and Taori et al. 2020 (synthetic-to-real divergence in image classification), Henderson et al. 2018 (mechanism claims in deep RL), and Pineau et al. 2021 (non-reproduction in published benchmarks).
> >
> > 2. The *conjunction* of these four failure modes occurring together in a single pipeline that nonetheless looked rigorous from the inside is the case study's contribution. We are not arguing the conjunction is common in published variant-calling work; we are arguing that the *structural pattern* — multiple individually-plausible methodology errors that compound — is a real and documented risk in applied ML, and that our case study illustrates how that pattern can produce a coherent-looking analysis that does not survive independent scrutiny.
> >
> > We acknowledge that a literature audit of recent ML-for-variant-calling papers would substantially strengthen this contribution. We have flagged this as an explicit limitation in §9 ("No literature audit") and described why we omitted it from this work. We believe the case study is publishable as a self-contained worked example of a methodology failure pattern documented elsewhere, but we agree it cannot do the heavier lifting of a field-wide survey.

---

> > > ### Author Response · Authors · 2026-06-07
> > > **Response to Reviewer TkK2 Comments (v3)**
> > >
> > > ### 6. Problem setup: binary classification vs. multi-class targets
> > >
> > > The reviewer is right that DeepVariant, Clair, and other modern callers use a multi-class genotype target (0/0, 0/1, 1/1), not a binary one. We have added an explicit paragraph in §2.1 ("Our simplified formulation"):
> > >
> > > > *"This is a strict simplification of the standard formulation, equivalent to collapsing the (0/1, 1/1) genotypes into a single positive class. We adopt this formulation only because the initial analysis we audit did so; we do not claim it is comparable to state-of-the-art callers. The methodology failures we document do not depend on this simplification: each could arise in a multi-class or pileup-tensor pipeline as well. The simplification does, however, limit the scope of conclusions we can draw about variant-calling performance: our model is not directly comparable to published callers in $F_1$."*
> > >
> > > The Limitations section §9 now also flags this directly under "Simplified binary classification target."
> > >
> > > ---
> > >
> > > ## Response to "Requested Changes"
> > >
> > > ### 7. Structure: Methodology / Experiment Setup / Analysis vs. four-finding sectioning
> > >
> > > We took this concern seriously. The reviewer's underlying worry — that the four-finding structure "risks giving the impression that experiments were arranged to support the stated findings, rather than that the findings emerged from a clearly defined experimental protocol" — is legitimate, and we have addressed it directly:
> > >
> > > - **New §2.7 "Experimental protocol across the four findings"** explicitly states: *"the four findings... were planned and executed as a single audit protocol, not assembled post-hoc to support a narrative. The protocol was: (1) train under each loss... (2) test the mechanism... (3) audit the feature pipeline... (4) re-implement the training loop... We adopted this protocol before running any of the four experiments. Each experiment was specified in advance; the findings report what we observed."*
> > > - The unified background and setup material is now consolidated in §2, with subsections for task formulation, architecture, losses, mixed-precision training, collapse definition, datasets, and the protocol itself. Each finding section (§3–§6) then has its own focused experimental design and results.
> > >
> > > We did not flatten the four findings into a single Methodology / Experiment / Analysis structure, because each finding answers a distinct question with a distinct experimental design (synthetic-vs-real ranking; controlled mechanism test; static and counterfactual leak audit; independent re-implementation). Reporting them as four separate experiments in the case study, with the protocol pre-stated in §2.7, seemed clearer than presenting them as analyses of a single experimental setup. We are open to reconsidering this if the reviewer feels §2.7 does not adequately address the original concern about post-hoc arrangement.
> > >
> > > ---
> > >
> > > We thank the reviewer again for a substantive review that improved the paper. The concerns raised have produced concrete revisions: a corrected scope (title, abstract, §1, §7.3, conclusion), an explicit framing of VariantCNN's relationship to prior work (§1, §2.2, §10), a justified formulation choice (§2.1, §9), a stated experimental protocol (§2.7), and tighter quantitative writing throughout.
> > >
> > > We hope the reviewer will reconsider their assessment of the paper in light of these changes.

---

### Review · Reviewer_yXKq · 2026-07-07

**Summary Of Contributions:**

The paper proposes a "methodology audit of one internally developed deep-learning pipeline for germline single-nucleotide variant calling, evaluated on the Genome in a Bottle.  Several alternatives are evaluated (comparison between binary cross entropy and focal loss, alternatives in architecture, variable fixed point representation). Founds results are analyzed

**Audience:**

No

**Audience Explanation:**

TMLR serves a broad machine learning audience rather than a highly specialized application community. Since I was selected as a reviewer based on my general machine learning background, I expected the paper to be accessible without requiring extensive prior expertise in genomics. However, I found that significant parts of the technical development rely on domain-specific concepts that are introduced only briefly or not sufficiently motivated. Consequently, I found it difficult to follow several aspects of the proposed methodology and to appreciate the claimed contributions. This suggests that the paper is written primarily for a narrow community with substantial prior knowledge of the application domain, which limits its accessibility and broader impact within the TMLR readership.

Into more detail:
- It is written with a very narrow audience in mind, and I am not part of that audience. As a result, I struggled to understand and follow significant portions of the paper. For example, I do not understand the relevance of the four claimed contributions listed on page 2.
- The experimental study is very narrow. The comparison between binary cross-entropy (BCE) and focal loss has been extensively studied, and I find it difficult to relate the reported findings in the genomic area to a broader research question. Other investigated aspects, such as variable representation precision, also concern well-established topics. The paper carefully frames them as findings in this experiment
- Although some observations regarding numerical precision may have broader relevance, the overall study is centered on a specific pipeline on a very specific problem. Moreover, the paper explicitly refutes alternative approaches, making it difficult to see how the conclusions generalize beyond the proposed setting.
- Overall, the paper reads more like a technical report produced in the context of a research project than a self-contained research article. It appears to present the conclusion of a longer investigation, but the motivation, intermediate steps, and broader context of that investigation are largely absent from the manuscript.

**Claims And Evidence:**

No

**Claims Explanation:**

For a more clear explanation please the "interest" section.

**Requested Changes:**

Again, I am not part of the target audience for this paper, so my assessment should be interpreted accordingly.

Nevertheless, a few observations can be made:

- None of the cited references is particularly recent. Moreover, the references published in major machine learning venues are fairly general and do not address the specific topic of this paper. The works mentioned by this paper as  related work on the same topic  have not been published in leading machine learning venues such as TMLR, NeurIPS, ICML, ICLR, AAAI,  TPAMI, etc. The cited works (Poplin et al., 2018; Luo et al., 2020; Zheng et al., 2022; Ramachandran et al., 2021; Zook et al., 2019) do belong to specific application domains.
- The paper lacks a dedicated discussion of related work that identifies a clear research gap and explains how the proposed work addresses that gap. As a result, it is difficult to assess the novelty and positioning of the contribution with respect to the existing literature.

---

> ### Author Response · Authors · 2026-07-20
> **Response to the Reviewer Comments V1**
>
> We thank the reviewer for their comments, and in particular for the repeated, fair-minded note that the assessment should be read in light of the reviewer's distance from the application area. The revised PDF makes concrete changes aimed squarely at that concern. We respond point by point below.
>
> We also note that this revision incorporates changes from the previous round in response to Reviewers TkK2 and iCjA, most relevant here being a restructuring so that the paper's constructive contribution, a four-item prospective protocol, now appears as Section 3, before the evidence, rather than near the end. Several of the points below build on that change.
>
> ---
>
> ## Accessibility: the paper reads as written for a narrow audience
>
> This was the reviewer's central concern and we have acted on it directly.
>
> **A self-contained primer for readers outside genomics.** Section 2 now opens with a paragraph titled "A minimal primer for readers outside genomics" that defines every domain term the paper uses (read, variant, reference, pileup, depth, allele frequency, truth VCF, high-confidence BED, GIAB) in five sentences, and then states explicitly that, with those definitions, the task is an ordinary supervised binary classification problem: each example is a genomic position, its features summarize the pileup, and its label indicates whether the position is a true variant. The paragraph ends: "No further genomics is required to read the paper." Our intent is that a reader with no genomics background can now read the paper end to end.
>
> **The four contributions are now motivated as general ML checks, not genomics findings.** The reviewer wrote that they did not understand the relevance of the four contributions on page 2. In the revision, the four experiments are no longer presented as standalone genomics findings. Section 3 states four prospective checks that make no reference to genomics: (1) synthetic-real consistency; (2) mechanism-controlled testing; (3) static label-leak audit; (4) independent-implementation reproduction. Sections 4 through 7 are then framed as one validation of each check. A new paragraph in Section 1, "Why these four checks," explains that these are the four load-bearing dependencies of the original conclusion, each tested once. The relevance of each contribution is now stated in general ML terms before any genomics appears.
>
> ---
>
> ## The experimental study is narrow; BCE-vs-Focal and precision are well-studied
>
> We want to clarify our claim, because we believe the reviewer read the paper as claiming novelty that we do not in fact claim.
>
> We do not claim novelty in the comparison of BCE and Focal Loss, nor in the study of numerical precision. These are, as the reviewer says, well-established topics. The paper's claim is the opposite of a novelty claim about these mechanisms: it is that a pipeline can invoke each of these well-understood tools correctly-looking and still reach a wrong conclusion, because the failure is in the methodology connecting them, not in any one of them. The Focal-vs-BCE comparison is not offered as a contribution; it is the setting in which the first failure (a synthetic benchmark that inverts the real-data ranking) is exhibited. Likewise the precision study is the setting for the second and fourth failures, not a contribution to mixed-precision training.
>
> We have revised the framing so this is unambiguous. The abstract, the introduction, and Section 3 now present the contribution as the four-check protocol and the documented conjunction of failures, with the loss functions and precision settings explicitly cast as the vehicle rather than the subject. Section 2.1 and the Limitations section state that our model is not competitive with published callers and is not intended to be.
>
> ---

---

> > ### Author Response · Authors · 2026-07-20
> > **Response to the Reviewer Comments V2**
> >
> > ## The study is centered on a specific pipeline and refutes alternatives, so generalization is unclear
> >
> > We do not claim that our specific failures generalize to other pipelines, and the revision states this explicitly in the title, the abstract, and in Section 7.3, which enumerates four boundaries on our claims. What we do claim to be general is the protocol, not the failures. The four checks in Section 3 are stated without reference to variant calling, to our architecture, or to our feature representation; each applies to any supervised pipeline that uses a synthetic benchmark, proposes a mechanism, engineers features from labelled data, or reports a headline empirical number. In that sense the generalizable output of the paper is the protocol, and it is deliberately decoupled from the specific setting in which we derived it.
> >
> > On "the paper explicitly refutes alternative approaches": we believe this refers to our reporting of negative results (that the mechanism hypothesis fails, that the precision effect does not reproduce). These refutations are internal to our own pipeline and are not claims about other approaches. We have clarified in Section 4.1 that our negative result about the gradient-noise mechanism concerns its applicability to our pipeline and does not bear on the validity of the general noise-as-regularization literature.
> >
> > ---
> >
> > ## The paper reads like a technical report, not a self-contained article
> >
> > We have restructured to address this. The paper now has an explicit logical spine: Section 3 states four checks; Sections 4 through 7 supply one experiment validating each check, in the same order; Section 8 discusses why standard practice missed each failure; and the Limitations and Related Work sections position the result. The motivation for studying exactly these four aspects, which the reviewer correctly noted was missing, is now given in Section 1 ("Why these four checks") as the enumeration of the original conclusion's load-bearing dependencies. Our aim was to convert what may indeed have read as the tail end of an internal investigation into a paper organized around a single, stated, general-purpose contribution.
> >
> > ---
> >
> > ## Requested change: references are not recent and not from leading ML venues
> >
> > We have added recent works from mainstream ML venues:
> >
> > * **D'Amour et al. (2022), JMLR**, on underspecification: ML pipelines routinely return many predictors with equivalent held-out performance but divergent behaviour under scrutiny, so held-out accuracy does not determine what a pipeline will do.
> > * **Bouthillier et al. (2021), MLSys**, on accounting for variance in ML benchmarks: variance from data sampling, initialization, and hyperparameters routinely swamps reported method gaps, which bears directly on our non-reproduction finding (Section 6).
> > * **Kapoor et al. (2024), Science Advances (REFORMS)**, on reporting standards for ML-based science, alongside the existing Pineau et al. (2021) reproducibility-program reference.
> >
> > These are now discussed in a dedicated positioning paragraph (see next point), not merely cited.
> >
> > ---
> >
> > ## Requested change: no dedicated related-work discussion identifying the research gap
> >
> > We have added this separately now. The Related Work section now opens with a paragraph titled "Positioning and research gap" that (a) summarizes the mainstream-ML methodology literature just cited, (b) states how our work sits within it, and (c) names the gap we fill. In brief: that literature studies failure mechanisms one at a time and in aggregate across many pipelines; our contribution is a worked, end-to-end account of how four independent failure modes compound within a single pipeline that nonetheless passed internal review, plus a minimal per-failure protocol derived from that account. We state explicitly that our four checks are complementary to, not substitutes for, the aggregate reporting standards of D'Amour et al., Bouthillier et al., and Kapoor et al.
> >
> > ---

---

> > > ### Author Response · Authors · 2026-07-20
> > > **Response to the Reviewer Comments V3**
> > >
> > > ## Summary of changes in this revision
> > >
> > > | Reviewer point | Change |
> > > |---|---|
> > > | Paper inaccessible to general ML readers | New "minimal primer for readers outside genomics" paragraph at the start of Section 2, defining every domain term and reframing the task as ordinary binary classification |
> > > | Relevance of the four contributions unclear | Contributions restated as four general-ML checks in Section 3 (moved forward from the end of the paper); new "Why these four checks" paragraph in Section 1 |
> > > | BCE-vs-Focal and precision are well-studied | Framing revised throughout to cast these as the vehicle, not the contribution; explicit statements that we claim no novelty in these mechanisms |
> > > | Generalization unclear; refutes alternatives | Section 7.3 boundaries retained; protocol reframed as the generalizable output, stated independently of the pipeline; Section 4.1 clarifies that the negative result is internal to our pipeline |
> > > | Reads like a technical report | Explicit logical spine (Section 3 checks to Sections 4 through 7 experiments); motivation added in Section 1 |
> > > | References not recent or from leading ML venues | Added D'Amour et al. (2022, JMLR), Bouthillier et al. (2021, MLSys), Kapoor et al. (2024, Science Advances) |
> > > | No related-work gap discussion | New "Positioning and research gap" paragraph opening the Related Work section |
> > >
> > > We thank the reviewer again.

---

### Review · Reviewer_iCjA · 2026-07-10

**Summary Of Contributions:**

The paper audits a specific, internally developed variant calling classifier model.
They study the loss-function dynamics of this model, in the aspects of failure cases such as synthetic vs real data, precision-collapse pattern, structural label leakage, and non-reproduction.
The paper introduces the details of how each failure happens and suggest minimal protocol to test these failures.

Strength:
- The paper presents its setup in detail.
- The paper clearly presents its findings.

Weakness:
- This paper is contributing to a narrow scope. And it (positions itself as an audition of a specific model) lacks a clear line of motivation and logic flow, but is more like an empirical report. I am not familiar with this form of paper, which makes me concerned about how much it could contribute to the audience.
- The paper focuses on one specific model architecture and training recipe, which makes me concerned about how much the findings could generalize.

**Audience:**

No

**Audience Explanation:**

- Firstly, this is an audit report for a specific model (maybe not mainstream) on a speicific tasks. As I personally have limited knowledge in this specific setup, I am unsure of the impact.
- Secondly, these findings are mainly failure modes, but without proposed solutions.
- Thirdly, these failure modes are mostly already there in other ML models and tasks (the author also points it out), so I am not very surprised that a similar failure is in there in this setting.

**Broader Impact Concerns:**

None.

**Claims And Evidence:**

Yes

**Claims Explanation:**

- Though the scope of the paper is narrow, the author is accurate in the claim or conclusion they made. For example, Prop. 4.1 is with detailed conditions on the settings.
- The claims are convincing to me because these failure modes are also there for other ML tasks or models. But it hurts the paper's novelty and contribution.
- Each findings have corresponding empirical results.

Question to authors:
- "The hypothesis that low-precision arithmetic acts as gradient noise that escapes the loss-landscape attractor." Can you explain more about the reference of the claim, whether it's from related literature, or it's newly made by you?

**Requested Changes:**

- I am not convinced why the authors studied specifically these four aspects in their model and their tasks. Is it something well motivated by the first principle, or are casual, side product findings during one model construction process?
- The paper's title in Openreview is not matched with the pdf.
- The paper is hard to follow as 1) weak connection between each section; 2) too many details in the main body, which distract the reading.

---

> ### Author Response · Authors · 2026-07-20
> **Response to the Reviewer**
>
> We thank the reviewer for a careful and fair review, and in particular for affirming that the claims are supported by accurate and convincing evidence. We also appreciate the specific, actionable nature of the requested changes. We have uploaded a revised PDF that acts on each of them.
>
> Before the point-by-point response, we note that this revision also incorporates changes made in response to Reviewer TkK2 in the previous round. Those changes are: a new title scoping the contribution to a single internal pipeline; a new paragraph in Section 2.1 justifying our binary-classification target relative to the multi-class genotype targets used by DeepVariant and Clair; a new paragraph in Section 2.2 stating that VariantCNN is not derived from any published caller; a new Section 7.3 enumerating explicitly what we do and do not claim; a citation for the variant-prevalence figure; a reframing of the Section 6.4 hypotheses so they are no longer ranked by plausibility; and tighter numerical referencing between prose and tables. Reviewer iCjA's copy of the PDF may predate some of these changes.
>
> ---
>
> ## Question: provenance of the gradient-noise hypothesis
>
> > *"The hypothesis that low-precision arithmetic acts as gradient noise that escapes the loss-landscape attractor." Can you explain more about the reference of the claim, whether it's from related literature, or it's newly made by you?*
>
> Thank you for asking. The hypothesis is from the existing literature, not newly made by us. We have added an explicit paragraph titled "Provenance of the hypothesis" at the start of Section 4.1.
>
> The general claim, that gradient noise acts as implicit regularization and can help optimization escape poor regions of the loss landscape, is developed by Smith et al. (2020) on the generalization benefit of SGD noise and by Wen et al. (2020) on structured covariance noise in SGD. Its extension to reduced-precision arithmetic specifically is discussed by Croci et al. (2022), who analyse stochastic rounding as a deliberate noise source, and is implicit in the numerical-precision training literature more broadly (Wang et al. 2018; Sun et al. 2019).
>
> What our initial analysis contributed was not the hypothesis but the inference that it explained the particular collapse pattern we had observed. It is that inference, and only that inference, which the experiment in Section 4 tests and rejects. The revision now states this explicitly, including the clarification that our negative result concerns the applicability of the mechanism to our pipeline and does not bear on the validity of the general noise-as-regularization literature.
>
> ---
>
> ## Requested change 1: why these four aspects specifically?
>
> > *I am not convinced why the authors studied specifically these four aspects in their model and their tasks. Is it something well motivated by the first principle, or are casual, side product findings during one model construction process?*
>
> We have added a paragraph titled "Why these four checks" to Section 1.
>
> The four experiments are not opportunistic findings. They are the four load-bearing dependencies of the initial analysis, in the sense that removing any one of them removes the basis for its headline conclusion. That conclusion ("Focal Loss is preferable to BCE for this task, and low precision confers a stability benefit") rested on exactly four things:
>
> * (a) a synthetic benchmark establishing the loss ranking,
> * (b) a real-data effect establishing that precision matters,
> * (c) a mechanism explaining why precision matters,
> * (d) a feature pipeline assumed to be free of label contamination.
>
> We enumerated these four dependencies before running any follow-up experiment, then designed one controlled test per dependency. The correspondence is one-to-one: Section 3 tests (a), Section 4 tests (c), Section 5 tests (d), Section 6 tests (b). Each of the four tests failed. Section 2.7 already stated that the protocol was fixed in advance of the experiments; the new Section 1 paragraph now explains the principle that determined which four experiments to run.

---

> > ### Author Response · Authors · 2026-07-20
> > **Response to the Reviewer Comments V2**
> >
> > ## Requested change 2: findings without proposed solutions
> >
> > > *Secondly, these findings are mainly failure modes, but without proposed solutions.*
> >
> > The paper did contain a proposed solution, a four-item prospective protocol, but it was buried at Section 8 after the Discussion, which we now recognise made it read as an afterthought rather than as the paper's constructive output. We have restructured accordingly.
> >
> > The protocol is now **Section 3**, immediately after Background and before any of the four experiments. It has also been expanded. Each item now carries an explicit "Catches:" clause naming the failure mode it targets and a worked statement of what it would have flagged in our own case. For example:
> >
> > > *"Static label-leak audit. [...] A mechanical version: rename the label variable throughout dataset construction, re-run the feature pipeline, and check whether any feature becomes undefined or changes value. Any feature that does was label-derived. Catches: label-derived features. In our case (Section 5) the renaming test would have raised a NameError on feature 6, identifying the leak in seconds, whereas ordinary code review did not."*
> >
> > We have also added two properties of the protocol that we think matter for how it should be used: each check is a purely negative test, so passing all four does not certify a result; and the four checks are independent, so they can be adopted individually. In our case each failure was caught by exactly one check.
> >
> > The abstract, introduction, and conclusion have been revised so that the protocol is presented as the paper's constructive contribution rather than as a closing remark, and the four experiments are framed as validations of the four checks.
> >
> > ---
> >
> > ## Requested change 3: title mismatch in OpenReview
> >
> > Thank you for the comment. The PDF title was updated in the previous revision round in response to Reviewer TkK2, and the OpenReview metadata was not updated to match. We have corrected the OpenReview title to match the PDF:
> >
> > > *"A Methodology Audit of a Single Deep-Learning Variant-Calling Pipeline: Four Reproducibility Failures in One Internal Classifier"*
> >
> > ---
> >
> > ## Requested change 4: weak connection between sections, too many details in the main body
> >
> > We have addressed the first part directly. Moving the protocol to Section 3 establishes a spine that the rest of the paper follows: Section 3 states four checks, Sections 4 through 7 report the four experiments that motivate them, one per check, in the same order. Section 3 now ends with an explicit statement of this correspondence, and Section 1 states it as well. Each finding section is now readable as the validation of a specific check rather than as an item on a list.
> >
> > On the density of the main body, we agree in principle and have not made large cuts in this round, because we were unsure which material the reviewer found distracting. Our own assessment is that the following are the strongest candidates for relegation to an appendix, and we would be glad to move any or all of them if the reviewer confirms:
> >
> > * The full feature-vector layout listing in Section 5.1, retaining only the leaking feature.
> > * The FP16 and BF16 bit-width and dynamic-range details in Section 2.4.
> > * The stochastic rounding pseudocode (Algorithm 1), retaining only the equations in the main body.
> > * The stage-by-stage structure of the leak audit in Section 5, compressible into a single paragraph plus the counterfactual table.
> >
> > If the reviewer would indicate which of these, or which other material, they found distracting, we will move it in the next revision.
> >
> > ---
> >
> > ## On scope and novelty
> >
> > **On narrow scope.** We agree that this is a case study of one pipeline, and the revised paper says so in the title, the abstract, the introduction, and in Section 7.3, which enumerates four explicit boundaries on our claims including that we have not audited DeepVariant, Clair, or any other published caller. We do not think this framing can be strengthened further without misrepresenting what we did. What we would offer in mitigation is that the protocol of Section 3 is not scoped to our pipeline: the four checks make no reference to variant calling, to our architecture, or to our feature representation, and are stated so that they can be applied to any supervised learning pipeline that uses a synthetic benchmark, proposes a mechanism, engineers features from labelled data, or reports a headline empirical result.
> >
> > **On novelty.** The reviewer observes that these failure modes already occur in other ML models and tasks, and that this hurts the paper's contribution. Our position is that the contribution is not the discovery of any individual failure mode but the documented conjunction of four of them in a single pipeline that passed internal review, together with the protocol derived from that conjunction.

---

> > > ### Author Response · Authors · 2026-07-20
> > > **Response to the Reviewer Comments V3**
> > >
> > > ## Summary of changes in this revision
> > >
> > > | Reviewer point | Change |
> > > |---|---|
> > > | Provenance of gradient-noise hypothesis | New "Provenance of the hypothesis" paragraph, Section 4.1, with four citations and an explicit statement that the hypothesis is from prior literature |
> > > | Why these four aspects | New "Why these four checks" paragraph, Section 1, with the four load-bearing dependencies enumerated and mapped one-to-one to Sections 3 through 6 |
> > > | No proposed solutions | Protocol promoted from Section 8 to Section 3; each item expanded with a "Catches:" clause and a worked statement of what it would have flagged in our case; abstract, introduction and conclusion revised to present it as the constructive output |
> > > | Title mismatch | OpenReview title corrected to match the PDF |
> > > | Weak connection between sections | Protocol at Section 3 now provides the organising spine; explicit one-to-one correspondence stated in Sections 1 and 3 |
> > > | Too many details in main body | Four specific candidates identified for relegation to appendix, pending reviewer guidance on which material was distracting |
> > >
> > > We thank the reviewer again for the review, and especially for the affirmative assessment of the evidence. We hope the restructuring around the protocol addresses the concern that the paper reads as an empirical report without a constructive line.